# Epigenetic Biomarkers as a New Diagnostic Tool in Bladder Cancer—From Early Detection to Prognosis

**DOI:** 10.3390/jcm13237159

**Published:** 2024-11-26

**Authors:** Natalia Jaszek, Alicja Bogdanowicz, Jan Siwiec, Radosław Starownik, Wojciech Kwaśniewski, Radosław Mlak

**Affiliations:** 1Department of Laboratory Diagnostics, Medical University of Lublin, 20-093 Lublin, Poland; alicja.bogdanowicz@umlub.pl (A.B.); radoslawmlak@umlub.pl (R.M.); 2Department of Pneumology, Oncology and Allergology, Medical University of Lublin, 20-090 Lublin, Poland; jan.siwiec@umlub.pl; 3Department of Urology and Urological Oncology, Medical University of Lublin, 20-090 Lublin, Poland; radoslaw.starownik@umlub.pl; 4Department of Oncological Gynaecology and Gynaecology, Medical University of Lublin, 20-081 Lublin, Poland; wojciech.kwasniewski@umlub.pl

**Keywords:** bladder cancer, biomarkers, epigenetics, early detection, prognosis

## Abstract

Bladder cancer (BC) currently ranks as the 9th most common cancer worldwide. It is characterised by very high rates of recurrence and metastasis. Most cases of BC are of urothelial origin, and due to its ability to penetrate muscle tissue, BC is divided into non-muscle-invasive BC (NMIBC) and muscle-invasive BC (MIBC). The current diagnosis of BC is still based primarily on invasive cystoscopy, which is an expensive and invasive method that carries a risk of various complications. Urine sediment cytology is often used as a complementary test, the biggest drawback of which is its very low sensitivity concerning the detection of BC at early stages, which is crucial for prompt implementation of appropriate treatment. Therefore, there is a great need to develop innovative diagnostic techniques that would enable early detection and accurate prognosis of BC. Great potential in this regard is shown by epigenetic changes, which are often possible to observe long before the onset of clinical symptoms of the disease. In addition, these changes can be detected in readily available biological material, such as urine or blood, indicating the possibility of constructing non-invasive diagnostic tests. Over the past few years, many studies have emerged using epigenetic alterations as novel diagnostic and prognostic biomarkers of BC. This review provides an update on promising diagnostic biomarkers for the detection and prognosis of BC based on epigenetic changes such as DNA methylation and expression levels of selected non-coding RNAs (ncRNAs), taking into account the latest literature data.

## 1. Introduction

### 1.1. Epidemiology

Currently, bladder cancer (BC) is positioned as the 9th most common cancer worldwide. In 2022, there were approximately 614,000 new cases of BC and 220,000 deaths attributed to the disease were recorded worldwide. BC is a particular issue for European populations. The highest incidence rate among women is observed in the Netherlands while the highest incidence rate among men is observed in Spain [1]. The occurrence of BC is primarily associated with external risk factors, and up to 50% of patients have the disease due to smoking tobacco products [2]. Other risk factors identified by the International Agency for Research on Cancer (IARC) include occupational exposure during the production of aluminium and rubber products, working with synthetic dyes, ingestion of opium, environmental exposure to X or gamma radiation and arsenic, taking drugs such as cyclophosphamide and chlornafazine, and infection with *Schistosoma* bacteria. Additionally, a higher risk of BC is observed in men than in women [2,3].

BC is characterised by high rates of recurrence and metastasis, which are primarily due to the late diagnosis of patients who are at an intermediate or advanced stage of this cancer, which is often due to the lack of clear symptoms of BC in its early stages or the presence of symptoms such as haematuria, which may be misinterpreted as other less serious disorders such as cystitis, urolithiasis, and others [4,5]. Crucial to prolonging the survival of patients struggling with BC are studies on tools for early detection, prediction of treatment effects, as well as prognostic techniques [6].

### 1.2. Etiology

Approximately 90% of BC cases are of urothelial origin, that is, originating in the epithelial tissue lining, among others, of the inner surface of the bladder. The remaining cases are squamous cell carcinoma, adenocarcinoma, or neuroendocrine carcinoma [7,8]. Due to the ability of the tumour to penetrate muscle tissue, BC is divided into muscle-invasive bladder cancer (MIBC) (stages T2–T4) and non-muscle-invasive bladder cancer (NMIBC), which includes approximately 75% (stages Tis, Ta, and T1) of all diagnosed cases [9].

The exact genetic basis of BC is not yet known and there are still many questions concerning its pathogenesis. It is also unclear whether specific genetic burdens have an impact on the increased risk of BC, and research is still underway to determine the exact genetic risk factors [9]. Deletion of chromosome 9 is a frequently observed genetic mutation in both NMIBC and MIBC. Due to this mutation, it is observed that there is a loss of the *CDKN2A* gene encoding two proteins: *p16* and *p14ARF*, which regulate the *RB* and *p53* pathways, and *TSC1*, which regulates *mTOR* signalling and is associated with an increase in the expression of *mTOR* molecular targets, such as telomerase reverse transcriptase *TERT*. An increase in the level of *TERT* plays a key role in maintaining telomere integrity and contributes to tumour progression in BC [10,11,12]. According to currently available information about the molecular characteristics of BC, point mutations within the *FGFR3* gene are known to occur in about 60% of patients diagnosed with NMIBC. Mutations within the *RAS* genes are also common but are mutually exclusive with mutations within *FGFR3*. Mutations in one or the other of these genes occur in about 90% of cases in stage Ta [9,10]. A study using human urothelial cells with mutated *FGFR3* showed increased growth, suggesting that it may play a very important role in the growth of cancerous tumours [13]. About 30% of cases also have mutations within the *PIK3CA* genes, often coexisting with *FGFR3* or *RAS* mutations, suggesting that NMIBC involves both activation of *RAS–MAPK* signalling as well as *PI3K* signalling [14]. Mutations in the *ERBB3* genes, leading to *PI3K* activation, are also observed at the T1 stage [15]. In NMIBC, there is also inactivation of some genes that encode chromatin regulatory proteins, such as *STAG2*, *KDM6A*, *KMT2D*, *KMD3C*, *CREBBP*, *EP300*, and *ARID1A*, but their function in BC development is not yet fully understood [14]. In the case of MIBC, the vast majority of cases have been found to involve the loss of cell cycle checkpoints due to mutations within *TP53*, *RB1*, or *ATM*, or due to changes in the activity of their regulators. In addition, the DNA damage response is impaired through the loss of *ATM* or *ERCC2* function [16,17]. As for activating *FGFR3* point mutations, they are less frequently observed than in NMIBC, even though increased expression of this gene is common [18]. In about 70% of MIBC cases, activation of the *RAS–MAPK* and *PI3K* pathways is observed, probably through positive regulation or mutation within the *ERBB2/3* genes [9,16]. In this type of BC, activation of *MET77* signalling and the *NOTCH* pathway as well as activation of the *AKT–mTOR* pathway are also observed [9,19,20].

The epigenetic mechanisms responsible for regulating the expression levels of selected genes in a manner unrelated to changes in the DNA nucleotide sequence also play a very important role in the pathogenesis of BC [10]. DNA methylation is a common epigenetic modification involving the regulation of gene expression through the methylation of CpG dinucleotides within the 5′ regulatory region of a gene. This reaction is catalysed by DNA methyltransferases. The methylation profiles of genes can be analysed both from so-called free circulating DNA (cfDNA) and in tumour cells excreted in urine [7]. Significantly, DNA methylation does not affect the continuity of base pairs within genes and is only observed in their promoter regions where it can affect protein–DNA interactions, thus regulating the expression of a given gene [21]. Molecules involved in epigenetic regulation of gene expression are, in addition, non-coding RNAs (ncRNAs), i.e., molecules that do not form a matrix for protein synthesis. These include microRNAs (miRNAs), long non-coding RNAs (lncRNAs), and circular RNAs (circRNAs). MiRNAs are a family of short RNAs of approximately 18–25 nucleotides in length that regulate gene expression at the post-transcriptional level by interacting with 3′ mRNA fragments that are not translated [22]. Through this action, miRNAs can inhibit mRNA translation, preventing the formation of protein products. In addition, these molecules can inhibit the translation of both suppressor genes and oncogenes, making their role in carcinogenesis very important [7]. Another type of ncRNAs are lncRNAs, which, unlike miRNAs, are long molecules consisting of more than 200 nucleotides [23]. These molecules play an important role in the epigenetic regulation of gene expression. Their mechanism of action varies widely and may be based, for example, on chromatin modification and remodelling, histone modification, changes in nucleosome positioning, or direct effects on the regulation of the expression of selected genes [7,23]. An interesting example of ncRNAs are circRNA molecules, which have a different structure in that they do not possess the 3′ and 5′ ends, which makes them resistant to degradation by RNA exonucleases [24,25]. It has been suggested that they play an important role in processes such as proliferation, invasion, migration, and apoptosis of cancer cells [26]. Moreover, epigenetic changes, such as DNA methylation, and expression levels of miRNAs, lncRNAs, and circRNAs can be easily detected in tissues and, most importantly, in readily available biological material such as blood and urine, often long before the clinical manifestations of cancer. Therefore, they are being investigated both to explain the molecular mechanisms involved in BC pathogenesis and as potential biomarkers and molecular targets for the development of modern diagnostic and therapeutic techniques [7,8].

### 1.3. Treatment

The treatment of patients with BC is usually multimodal and can combine surgery, chemotherapy, radiation therapy, and targeted therapy including immunotherapy [9]. A frequently used diagnostic method and, in the case of NMIBC, a therapeutic method, is transurethral resection of the bladder tumour (TURBT). It allows determining the stage of the tumor and, in the case of NMIBC, allows complete resection of the tumour [9,27]. TURBT is the primary standard of care for NMIBC, both in combination with and without single dose intravesical chemotherapy, but it is a major procedure and can place a heavy burden on the patient [9]. In the case of NMIBC, immunotherapy with the BCG vaccine administered intravesically is also a common treatment [28]. It is used to stimulate the inflammatory response and activate macrophages. However, this therapy is associated with the risk of side effects such as bladder and prostate inflammation/infection, sepsis, fever, and general malaise [29]. For patients intolerant of BCG therapy, intravesical maintenance chemotherapy is also being introduced, leading to a reduction of about 44% in recurrences occurring at one year [30]. For MIBC and NMIBC patients in whom BCG treatment has failed to elicit a response, the standard of care is the so-called radical cystectomy [31]. In men, it involves removal of the bladder, prostate gland, and seminal vesicles and distal ureters, while in women, it involves removal of the bladder, entire urethra, anterior vaginal wall, uterus, and distal ureteral segments [31]. An alternative treatment for MIBC is trimodal therapy (TMT), which combines TURBT followed by simultaneous radio- and chemotherapy. TMT allows for bladder preservation and is another option for patients ineligible for radical cystectomy. In addition, for some patients, partial cystectomy is suggested as a less burdensome method. This is mainly true for patients with lower grade MIBC [31,32]. In the 1990s, combination chemotherapy based on cisplatin for metastatic BC became the mainstay of BC treatment [9]. Although it is the standard of care for BC, not all patients qualify for its use, so it is often replaced by gemcitabine in combination with carboplatin [33]. Additionally, in recent years, several new treatment strategies have been proposed to be incorporated into standard treatments. The first is treatment with erdafitinibum, a small-molecule inhibitor of fibroblast growth factor 3, which has seen a response rate of 42% in phase 2 trials in patients previously treated with platinum-based chemotherapy [34]. The next new therapy is based on an antibody-drug conjugate called enfortumab venodine, which is directed against nectin-4 [35,36]. Another antibody-drug conjugate is sacituzumab govitecan, in which a monoclonal antibody targets TROP2 combined with a topoisomerase I inhibitor. In this case, the response rate was 27% in patients previously treated with platinum-based chemotherapy [37]. It is worth mentioning that immunotherapy is an important aspect in the treatment of BC, but due to the difficulties associated with the immunosuppressive tumour microenvironment, the effectiveness of immune checkpoint inhibitors is significantly limited [38]. Therefore, it seems necessary to develop new independent biomarkers based on integrated genomic analysis to detect BC at a much earlier stage of progression, better predict clinical outcomes, and tailor treatment to patients at high risk of progression [39].

Innovations in medicine are developing at an extremely fast pace. However, it should be emphasised that many of the new developments are still in the phase of verification studies that will allow to assess their effectiveness and safety. In particular, epigenetic biomarkers are becoming an increasingly promising tool in predicting the course of disease and early diagnosis [40]. They have wide potential as multifunctional indicators to revolutionise treatment modalities and personalise therapy for patients with bladder cancer, which is crucial in the development of modern medicine [41].

### 1.4. Prognosis

A crucial issue from the perspective of clinical practice is the accurate assessment of the survival chances of patients diagnosed with BC. Properly determined patient prognosis is necessary to select the best possible individualised treatment regimen that is optimal for a given patient. Determining the chance of survival is based primarily on an assessment of the stage of the tumor, the size of the cancerous tumour, the type of BC (NMIBC or MIBC), the patient’s demographic and clinical factors, and the patient’s overall health. Statistics published by the National Cancer Institute (NCI) indicate that the 5-year relative survival rate for BC in situ is 91%, for localised BC it is 71%, for BC that has spread to nearby lymph nodes and organs it is 39%, and for metastatic BC it is 8% [42]. Many of the currently conducted analyses focus on the search for new biomarkers that could provide accurate prognostic indicators, given that the currently used criteria, due to the heterogeneity of BC, genomic instability, and differences in sensitivity to chemotherapy or immunotherapy, are not effective enough [43,44].

The following section will provide a review of selected recent literature reports indicating epigenetic changes with high potential for use as biomarkers for BC detection and a summary of the diagnostic tests developed.

## 2. Methodology

This paper is based on a literature review of epigenetic biomarkers for bladder cancer focusing on the potential usefulness of selected alterations for diagnosis and prognosis in BC. This narrative review focuses primarily on data on epigenetic alterations, such as methylation levels of specific genes or combinations of genes, and expression levels of ncRNA molecules, such as microRNAs, lncRNAs, and circRNAs.

Articles were searched in databases such as “Pubmed” and “Google Scholar” in order to access scientific papers related to the topics covered. The search for relevant publications was based on several keywords and their combinations. The main keywords used in the search included “bladder cancer”, “biomarkers”, “epigenetics”, “early detection”, “prognosis”, and “epigenetic biomarkers”. The selection of relevant articles focused on works published in the last eight years. The articles included in the bibliography published earlier than 2016 were used for the introductory section, where the current state of knowledge on the epidemiology, etiology, treatment, prognosis, and issues concerning the classical approach to the diagnosis of BC was described. The main goal was to collect the latest discoveries in the field on the subjects under discussion and update the state of knowledge and, therefore, the focus was particularly on results published in recent years. In order to perform this review, we looked for papers that included an assessment of the diagnostic usefulness of epigenetic biomarkers or their combinations. As for biomarkers used for BC detection, indicators such as sensitivity and specificity were chosen (with only biomarkers with sensitivity and specificity of no less than 60% being considered). As for biomarkers with potential for prognostic use, we chose articles with prognostic indicators such as OS (overall survival), PFS (progression-free survival), CSS (cancer-specific survival), DSS (disease-specific survival), or RFS (recurrence-free survival). Publications that did not include the aforementioned data of diagnostic or prognostic utility were not included in this review.

The selected scientific papers were analysed, and the data contained within them were grouped to clearly and transparently present potential epigenetic biomarkers of BC and their diagnostic and/or prognostic role. Through such analysis, it was possible to draw conclusions about the latest trends and most promising epigenetic changes that could be used as novel diagnostic and prognostic tools in BC patients.

## 3. Diagnosis of Bladder Cancer

### 3.1. Classic Approach

Currently, invasive cystoscopy remains the diagnostic gold standard for BC, but in addition to the heavy burden it carries, it is also a highly expensive method. Standard white light cystoscopy (WLC) is an inaccurate method, so improvements have begun to be made using techniques such as laser-induced fluorescence (LIF), autofluorescence cystoscopy (AFC), and photodynamic diagnosis (PDD). PDD involves infusing a photosensitiser into the bladder prior to cystoscopy, which causes cancer cells to exhibit red fluorescence in the presence of blue light [45]. Additionally, a 2021 meta-analysis by Veeratterapillay et al. showed that the use of PDD increased recurrence-free survival (RFS) in patients with NMIBC compared to WLC [46]. Urine sediment cytology is also frequently used as a complementary test, demonstrating a sensitivity and specificity of 48% and 86%, respectively. However, despite a fairly high sensitivity for high grade tumours of 84%, the sensitivity for low grade tumours was only 16% [47]. However, the results of urine cytology are affected by the presence of other disorders related to bladder function, such as inflammation, urinary tract infection, or urolithiasis [48]. To monitor BC, optical biopsy techniques such as optical coherence tomography (OCT) or confocal laser endomicroscopy (CLE) are also used. Combinations of imaging with other diagnostic methods give high sensitivity and specificity values, such as 89.7% and 100%, respectively, when PDD is combined with OCT [45,49]. Ultrasound examinations are also used to diagnose BC. An example of a relatively effective technique is high-resolution 29 mhz micro-ultrasound (mUS), which shows high sensitivity and specificity for BC with low malignancy (85% and 89%, respectively) [45,50]. Despite continuous improvements in BC diagnostic techniques, there are still many cases of this disease detected at a late stage. Diagnosis of BC is expensive and, among the diagnostic methods, invasive cystoscopy is still the most commonly used, which carries a high burden for patients involving the possibility of complications such as painful urination, bleeding, urinary tract infections, fever, or urinary retention [51]. Therefore, a search is in progress for new diagnostic methods based on biomarkers that could significantly reduce the costs of both initial diagnosis and health monitoring of patients and could enable non-invasive detection of BC even at low stages of advancement.

### 3.2. Epigenetic Biomarkers in the Diagnosis of BC

The use of epigenetic alterations as biomarkers for the early detection of BC has been the subject of intense research in recent times. Currently, many potential biomarkers based on the modifications in question are already known, but still, few of them have sufficient sensitivity and specificity, especially when it comes to the detection of NMIBC, or have high costs or complicated procedures for their analysis. Therefore, many issues regarding the published findings are still in question, and there is a great need to search for new high-specificity and high-sensitivity epigenetic alterations that could, in the future, provide a basis for the diagnosis of the disease entity in question and help reduce the need for invasive cystoscopy, which is still the gold standard for BC diagnosis. In this review, the authors summarise selected epigenetic biomarkers, including changes in the methylation levels of selected genes and expression of selected ncRNAs undergoing validation studies, and compare the potential for their use, including analysis of parameters such as sensitivity (SN) and specificity (SP). In addition, discussed are selected epigenetic changes showing high potential as BC diagnostic biomarkers that are in the phase of research using BC cell lines or have been selected on the basis of available bioinformatic analyses.

### 3.3. Available Tests Based on Analysis of Epigenetic Changes for BC Diagnosis

Several tests based on the methylation of selected DNA fragments have already been developed so far and investigated for use in BC diagnosis. A number of diagnostic panels based solely on DNA methylation have now been developed. The first of these is Bladder EpiCheck, which is a commercially available test. The diagnostic panel consists of the DNA methylation in 15 genes, which uses qPCR to assess their methylation levels (SN = 68.2%, SP = 88%) [48,52]. The next test is UroMark, which shows very high sensitivity and specificity (98% and 97%, respectively). It is based on the analysis of 150 methylation sites using next-generation DNA sequencing (NGS) in which the genetic material has undergone bisulfide conversion [48,53]. Another test is BladMetrix, which evaluates the methylation levels of eight selected genes using droplet digital PCR (ddPCR). In a study involving patients with haematuria, the sensitivity and specificity of the test were 92.1% and 93.3%, respectively [48,54]. A 2021 clinical evaluation of the Bladder CARE test by Piatti et al. showed its overall sensitivity and specificity to be 93.5% and 92.6%, respectively. For the determination, it uses urine samples in which the analysis of three methylated BC-specific genes is performed: *TRNA-Cys*, *SIM2*, and *NKX1*. The analysis is performed using methylation-sensitive restriction enzymes [55]. Also published in 2024 was a paper by Bang et al. presenting the results of a diagnostic evaluation of the EarlyTest BCD test that analysed the methylation level of the *PENK* gene in samples of excreted urine. This study revealed the high potential of the test, given that the overall sensitivity and specificity were 81% and 91.5%, respectively, in detecting all stages of BC in patients with haematuria. Sensitivity was significantly related to the stage of BC [56]. A surprisingly positive result was shown by the use of the GynTect test, originally designed for use in the diagnosis of cervical cancer. It is based on the use of six DNA methylation markers (ASTN1, DLX1, ITGA4, RXFP3, SOX17, and ZNF671), and the sensitivity and specificity of the test were 60% and 96.7%, respectively, noting that the material was collected only from patients with NMIBC [57]. A summary of the aforementioned tests can be found in Table 1.

In most of these studies, the level of methylation was assessed using biological material such as exfoliated cells found in urine. The discussed tests showed promising results in terms of their implementation in routine diagnostic testing in BC. However, the problems included often too low sensitivity, especially for BC with low malignancy, complicated testing procedures, or high costs associated with the use of these assays in practice. These results, however, have still not translated into the practical use of test molecules or modifications in routine diagnostic procedures. Several diagnostic tests approved by the Food and Drug Administration (FDA) based on urine samples are currently available, but none of them are based on the epigenetic changes discussed in the articles. Among them are four tests based on protein biomarkers and a test based on the examination of exfoliated cells. Protein biomarkers include BTA (bladder tumour antigen) on the basis of which two diagnostic tests have been constructed. Qualitative BTA-Stat is a rapid immunochromatographic test (sensitivity and specificity of 64% and 77%, respectively) and BTA-TRAK is an ELISA immunoenzymatic test (sensitivity and specificity of 65% and 74%, respectively). Both tests detect and measure the level of human complement factor H-related protein (hCFHrp) in urine samples. Another FDA-approved BC protein biomarker is NMP22 (nuclear matrix protein 22) on the basis of which two diagnostic tests have also been developed: the quantitative NMP22 ELISA (sensitivity and specificity of 69% and 77%, respectively) and the qualitative NMP22 BladderChek test (sensitivity and specificity of 58% and 88%, respectively), which have been validated for initial diagnosis. Also commercially available is the UroVysion test, which uses a multi-probe FISH (fluorescence in situ hybridisation) technique to detect four chromosomal abnormalities that have been linked to the development of BC: aneuploidy of chromosomes 3, 7, and 7, and loss of the 9p21 locus. UroVysion achieved a specificity of 87.7% and a sensitivity of 63%. Although the aforementioned tests have been approved by the FDA, due to their limitations, they are not widely used in clinical practice [48,58].

### 3.4. Epigenetic Biomarkers in BC—Cohort Studies

Recently, there have also been the results of studies evaluating the usefulness of analysing the methylation levels of single genes or panels of a few selected genes using material collected from BC patients such as blood, urine, or tumour tissues. Most of them, despite surprisingly positive sensitivity and specificity values, require further studies and proper clinical validation. The 2022 methylation panel of three genes developed by Fang et al., including *PCDH17*, *POU4F2*, and *PENK* hypermethylation, showed a sensitivity of 87% and a specificity of 97%. The biomarkers were tested using a sediment of cells present in urine. However, in the present study, it was not possible to determine the sensitivity and specificity in early and late stages of the disease due to the study involving too few patients and missing data. In addition, in the conducted study, the methylation panel did not show the potential to distinguish between different subtypes of BC [58,59]. In 2022, Hentschel et al. showed high potential for the combined use of methylated *GHSR* and *MAL* genes with a diagnostic potential that reached a sensitivity of 80% and a specificity of 93% [60]. Jiang et al. in 2024 also published a paper evaluating the utility of using a panel based on assessing the hypermethylation of three genes, *ZNF671*, *OTX1*, and *IRF8*, which were tested in patients’ excreted urine samples. The panel achieved a specificity of 90.9% and a sensitivity of 75% [61]. Another panel studied by Wu et al. consisting of four hypermethylated genes, *ONECUT2+*, *HOXA9*, *PCDH17*, and *POU4F2*, showed a specificity of 73.2% and a sensitivity of 90.5% in urine samples from patients with haematuria [58,62]. In 2020, Chen et al. evaluated the diagnostic utility of a methylation panel of two BC-specific genes (OTX1 and SOX1-OT) in the DNA of exfoliated cells found in urine. The sensitivity and specificity values of this panel were 91.7% and 77.3%, respectively [48,63]. Another publication in 2024 by Zhang et al. showed sensitivity and specificity values of 78% and 83%, respectively, for distinguishing patients with BC of urothelial origin from healthy volunteers using methylation analysis of *TWIST1* and *VIM* gene promoters in the deposits of cells contained in urine [64]. It is worth noting that the following review considers only selected markers based on DNA methylation that could be useful in the diagnosis of BC, while most of them still show many limitations due to, among other things, too few patients studied and require further approaches for their clinical validation.

A summary of the discussed DNA methylation-based biomarkers and proposed biomarkers not mentioned in the text is provided in Table 2 and Figure 1.

Changes in the expression levels of non-coding RNA (ncRNA) molecules are another epigenetic mechanism within which BC diagnostic biomarkers are sought. One of the types of ncRNAs considered in this review are miRNA molecules. An enormous amount of data has been published in recent years suggesting that miRNAs have great diagnostic potential in BC. A 2021 paper by Wang et al. analysed the diagnostic potential of miRNAs. It showed that three molecules detected in blood, miR-20a-5p, miR-92a-3P, and miR-17-5p, the expression levels of which were increased in BC, had a sensitivity of 90.4% and a specificity of 94.4% [71,72]. On the other hand, the results of the analysis by Yu et al. in 2023 of the expression levels of three miRNA molecules (miR-27b, miR-381-3p, and miR-451a) showed high values of sensitivity and specificity (86.9% and 77.38%, respectively) in detecting BC. These molecules are detected in blood serum, are associated with the regulation of the expression of genes such as *SMAD4* and *FOXO1*, and appear to show great potential as non-invasive biomarkers for the early detection of BC [73]. A broad spectrum of miRNA molecules is detected in urine and they often have high sensitivity and specificity. For example, the analysis of the expression levels of two molecules, downregulated miR-145 and upregulated miR-182, by Suarez-Cabrera et al. was characterised by sensitivity and specificity values of 93% and 86%, respectively [74]. In 2021, El-Shal et al. reported high sensitivity and specificity of a panel comprising two molecules: miR-96-5p and miR-183-5p (88.2% and 87.8%, respectively). The researchers suggested that these miRNA molecules could serve as potential novel diagnostic biomarkers for BC [75]. Other scientific studies have indicated that these molecules are involved in the regulation of transcription factors from the *FOXO* family, which function as tumour suppressors. Additionally, miR-183-5p directly promotes *PDCD4*, a pro-apoptotic molecule, whereas miR-96 has been shown to act as a tumour suppressor in pancreatic cancer by regulating the oncogene *KRAS* [76,77,78]. Relatively high diagnostic performance parameters were observed by Lin et al. for two molecules, miR-93-5p and miR-516-5p, with enhanced expression in BC. The molecular target of these miRNAs is the regulation of *BTG2*. In this case, miRNAs located in extracellular vesicles found in urine were studied and, combined together, they achieved sensitivity and specificity values of 85.2% and 82.4%, respectively [72,79]. In a 2017 study, Matsuzaki et al. reported the high diagnostic potential of miR-21-5p (SN = 75%, SP= 98%) for early detection of BC [80]. In 2019, Chen assessed the diagnostic potential of miR-101 as a biomarker to distinguish patients with BC from healthy volunteers. This potential biomarker showed a relatively satisfactory sensitivity and specificity of 82% and 80.9%, respectively [81]. In 2024, Lu et al. evaluated the diagnostic potential of six miRNAs (miR-221-5P, miR-181a-5p, miR98-5p, miR-15a-5p, miR-222-3p, and miR-197-3p). Among them, a panel of four molecules (miR-221-5P, miR-181a-5p, miR-15a-5p, and miR-222-3p) showed sensitivity and specificity values of 82.14% and 85.71%, respectively [82]. Currently, a huge number of studies exploring the potential of miRNA molecules as diagnostic biomarkers of BC are emerging. However, most of them, despite promising sensitivity and specificity results, require further study due to various limitations of the experiments performed, as the results are often not very reproducible [83]. Despite this, the currently available literature reports clearly indicate the definite diagnostic potential of miRNA molecules.

Another type of ncRNAs that is promising for BC detection is lncRNAs. In a comparative analysis of seven publications providing specificity and sensitivity results of the *UCA1* molecule, it was found that it achieved a sensitivity of 84% and specificity of 87% [48,84]. On the other hand, a diagnostic panel consisting of three different lncRNAs (*MALAT1*, *PCAT-1*, and *SPRY4-IT1*) showed a specificity of 85.6% and a sensitivity of 62.5% [85]. Another panel, combining four lncRNAs (*UCA1-201*, *HOTAIR*, *HYMA1*, and *MALAT1*) showed a sensitivity of 93.3% and specificity of 96.7% and was able to effectively distinguish NMIBC patients from BC patients [86]. In 2024, Gao et al. investigated the diagnostic potential of the hypermethylated RMRP gene for BC diagnostics. Their study demonstrated that it exhibited high diagnostic potential when using the RT-qPCR method, with a sensitivity and specificity of 83% and 70%, respectively. However, the most intriguing results were obtained using the modern RT-RAA-CRISPR/Cas12a method, which showed a surprisingly high diagnostic potential for this epigenetic modification, with a sensitivity and specificity of 95% and 92.5%, respectively [87]. Additionally, Liu et al. demonstrated that hypermethylation of the TERC gene also shows promising diagnostic potential, with reported sensitivity and specificity levels of 78.7% and 77.8%, respectively [88].

Other ncRNAs studied for their potential use as diagnostic biomarkers for BC include circRNAs. In 2024, Yang et al. demonstrated the high diagnostic potential of circRNA0071196, with a sensitivity and specificity of 87.5% and 85%, respectively. This circRNA is likely involved in regulating miR-19b-3p and CIT expression [89]. Another interesting example is hsa-circ0137439, which exhibited a diagnostic potential to distinguish BC patients from controls, with a sensitivity of 87.9% and specificity of 80.1%. For distinguishing NMIBC patients from MIBC patients, the sensitivity was 88.6% and the specificity was 73.5% [90].

A summary of the ncRNA-based biomarkers discussed and proposals for biomarkers not mentioned in the text are provided in Table 3 and Figure 1.

Despite the promising results and the high potential offered by epigenetic alterations, most of the proposed biomarkers still require further research. None of the available biomarkers have found application in clinical practice yet due to the lack of adequate validation, the high cost of developed tests and diagnostic panels, or complicated and expensive analytical procedures. Some of the proposed biomarkers show too low sensitivity; additionally, they do not allow, at this point, the detection of cancer at an early stage, which is key to increasing the chances of patient survival and accelerating the selection of appropriate treatment. Therefore, there is still a need to search for new highly sensitive and highly specific biomarkers for early detection of BC.

### 3.5. Potential Epigenetic Biomarkers in BC (Bioinformatic Analyses and Cell Line Studies)

The following section presents examples of proposed diagnostic biomarkers for BC. It includes both selected results of recent bioinformatic analyses and results of studies conducted on cell lines, suggesting possible development paths and potential new subjects for future research using biological material such as readily available urine and blood.

In the year 2023, a bioinformatic analysis of potential biomarkers related to BC diagnosis based mainly on data from the Gene Expression Omnibus (GEO) database was published. The analysis included selected genes hypermethylated in BC for which their expression was downregulated following this epigenetic modification [105]. One of them was *SPARCL1*, which belongs to the acidic and cysteine-rich protein family secreted in the cellular matrix and is responsible for encoding an extracellular matrix glycoprotein. It is involved in the development and progression of various cancers. Significant downregulation of gene expression is observed in BC, breast, cervical, rectal, lung, and ovarian cancers, among others, and it is considered a tumour suppressor gene [106]. In addition, the product of the *SPARCL1* gene affects the viability of cancer cells and exhibits anti-adhesion effects [107,108,109]. Therefore, *SPARCL1* methylation shows great potential as a biomarker for BC detection and a prognostic biomarker [105,106]. The above-mentioned bioinformatic analysis also indicated a significant difference in the methylation level of the *MFAP4* gene, which encodes an extracellular matrix protein. In comparison between healthy and BC tissues, the expression of *MFAP4* gene was reduced in BC. *COX7A1* was also mentioned as a potential diagnostic biomarker in BC. This gene encodes the 7A1 subunit of cytochrome c oxidase, which plays a key role in the mitochondrial electron transport chain. Its role as a useful biomarker in cancer prognosis, such as in gastric cancer, is now suggested [110]. The bioinformatic analysis mentioned earlier also suggested the potential of using the methylation level of the *EFEMP1* gene as a potential BC diagnostic biomarker [105]. It encodes the epidermal growth factor-containing fibulin-like extracellular matrix protein 1, but its function in tumorigenesis is not yet fully understood. Additionally, it shows high sensitivity and specificity (77.8% and 97.3%, respectively) for prostate cancer diagnosis [111]. The analysis also suggested the potential of using *ABCA8* and *MAMDC2* gene methylation as potential BC diagnostic biomarkers, but as with the methylation patterns mentioned earlier, they require further analysis [105].

Among potential BC diagnostic biomarkers, there are also several emerging studies using BC cell lines to analyse expression levels and the role that miRNAs play in BC pathogenesis. An interesting biomarker candidate is the miR-3622 molecule, which is a negative regulator of *LASS2*, known as a tumour metastasis suppressor gene. Overexpression of miR-3622 was observed to lead to increased proliferation and invasion of BC cells [112]. In 2022, Xu et al. conducted an analysis of miR-494-3p expression using BC cell lines and human bladder epithelial cells in which they showed that the expression level of this molecule was increased in BC cells. In addition, it was determined that it promoted the growth of BC cells by regulating the *KLF9/RGS* axis, suggesting that it may be an interesting object of study using patient-derived biological material to analyse its potential use as a diagnostic biomarker or therapeutic target in BC [113]. Another miRNA molecule is miR-616-5p, the expression level of which in BC cell lines was studied in 2021 by Ren et al. It was proven to promote the proliferation of BC cells in which its expression level was increased [114].

Currently, numerous studies are focused on the role that lncRNAs play in both cancer diagnosis and therapy. A recent study conducted on cell lines also suggested the exploratory potential of the lncRNA *DUXAP10* [115]. This experiment showed that the expression of *DUXAP10* in human BC cell lines was significantly higher than in non-BC cell lines. After knockdown of the *DUXAP10* gene, it was found that BC cell proliferation was inhibited, whereas in normal cells, the process was not significantly affected. In addition, a significant reduction in Akt and mTOR phosphorylation involved in the PI3K/Akt/mTOR signalling pathway, which is critical for cancer, was observed [116]. Such results provide hope not only for the use of assessing lncRNA *DUXAP10* expression levels as a diagnostic or prognostic biomarker in BC but also for the development of novel therapeutic approaches for this disease. In 2024, a paper by Arim et al. was published showing the results of a study of the diagnostic potential of lncRNA *BCYRN1*. A knockdown was performed, which resulted in a decrease in BC cell viability. In addition, the serum levels of *BCYRN1* in BC patients and healthy subjects were also examined, showing that its levels were significantly higher in BC patients. Such results suggest the possibility of the use of this lncRNA both as an epigenetic biomarker for the diagnosis of BC as well as a potential molecular target for therapy of this disease [117]. Another potential lncRNA-based biomarker is upregulated *XIST*, the molecular targets of which include *TET1* and the p53 protein, where it negatively regulates the p53 protein by binding to *TET1*. *XIST* plays an important role in regulating proliferation, migration, and apoptosis in BC cells [118,119]. In 2024, Zhou et al. reported the possible potential of *GAS6-AS1* as a novel biomarker useful in the diagnosis, prognosis, and treatment of BC. The researchers investigated the mechanism of action of this lncRNA and indicated that it was involved in an interaction with miR-367-3p and *PRC1*. *GAS6-AS1* was found to increase *MMP7* expression through interaction with miR-367-3p and decreased *APC* expression through *PRC1* binding, thereby promoting BC progression [120].

Nowadays, many scientific reports attribute an important role in BC pathogenesis to circRNA molecules. Given this, many researchers are looking for potential diagnostic biomarkers among them. In 2022, Yang et al. investigated the expression level of Hsa_circRNA_0088036 using BC cell lines and tissues from BC patients and proved that the expression level in BC cells was significantly higher compared to that in healthy cells. Knockdown of Hsa_circRNA_0088036 led to the inhibition of growth, migration, and invasion of BC cells. The increased expression of this molecule suggested the possibility of using it as a diagnostic biomarker of BC [121]. In 2021, the molecular mechanism of action and expression levels of circFLNA were analysed. It was found that this molecule played an important role in inhibiting BC cellular grounding by regulating miR-216a-3p and *BTG2*, and its level in BC cells and tissues was downregulated [122]. In 2024, Yi et al. identified a novel circPSMA7 molecule that showed increased expression levels in both BC cell lines and tissues. Additionally, it was found to promote cell proliferation and metastasis formation, playing a key role in tumour progression. Therefore, the authors suggested that circPSMA7 may be considered a potential diagnostic biomarker and additionally may represent a novel therapeutic target in BC patients [123].

## 4. Epigenetic Biomarkers in Clinical Prognosis of BC

### 4.1. DNA Methylation Biomarkers—Cohort Studies

There are many reports indicating the potential of DNA methylation biomarkers in predicting the course of cancers as well as the response to the treatment used [124]. An example is the study by Yoon HY. et al. in 2016, which looked at *RSPH9* methylation in BC. They indicated that hypermethylation of this gene was an independent predictor of relapse and disease progression, as evidenced by hazard ratios (HR) of 3.02 (*p* = 0.001) and 8.25 (*p* = 0.028), respectively [125]. A similar analysis was performed by Zhan L. et al. in 2017, who showed that hypermethylation of the *RASSF1A* gene was significantly more common in patients with BC and was associated with a shortened OS (HR = 2.24, 95% CI: 1.45; 3.48; *p =* 0.0003) [126]. On the other hand, Shivakumar M. et al. in 2017 observed a correlation between hypomethylation of the *NACC2* gene and OS (HR not reported; *p* = 0.0321). *NACC2* contributes to the inhibition of *MDM2* expression, which is responsible for normal p53 expression, leading to BC progression and shorter patient survival [127]. Analogous results were reported by Zhang Y. et al. in 2018, who showed that hypomethylation of the *CDH1* gene was associated with the occurrence of reduced OS in BC patients (HR not reported; *p* > 0.0001). The protein encoded by *CDH1* is closely associated with the regulation of cell division, and hypomethylation of this gene results in excessive cellular proliferation and BC progression [128].

Recently, there have been other new biomarkers based on DNA methylation crucial in predicting the course of the disease as well as the survival of BC patients [129]. An analysis by Takagi K. et al. in 2022 indicated that *CALN1* hypomethylation was correlated with advanced tumour stage (rho = −0.563, *p =* 0.0007) and was an independent risk factor for recurrence, as suggested by an HR of 3.83 (95% CI; 1.14–13.0, *p =* 0.031) [130]. Analogous results in peripheral blood samples were obtained by Chen JQ. et al. in 2022, as they observed a statistically significant relationship between reduced methylation levels of the *BLCAP* gene and cancer recurrence or death within 10 years (*p* = 0.03; false discovery rate (FDR) = 2.63 × 10^−14^) [131]. A similar study was conducted by Zhang C. et al. in 2020 to assess the predictive value of methylation of certain genes, including *KRT8*. Their results showed that a hypomethylated *KRT8* gene was correlated closely with the prognosis of survival of patients with bladder cancer (*p =* 0.003), which was also confirmed by an HR of 6.74 [132]. Kim C. et al. also focused on the relationship between a certain level of DNA methylation and the prognosis of patients with BC and observed that hypermethylation of the *PTK2* gene was associated with a statistically significant increased mortality (HR not reported, *p* = 0.022). In addition, increased *PTK2* expression also contributed to poor survival outcomes in this group of patients [133]. However, DNA methylation-based biomarkers, despite their potential use in bladder cancer prognosis, still require further study [134].

A summary of the discussed DNA methylation-based biomarkers and proposed biomarkers not mentioned in the text are provided in Table 4 and Figure 2.

### 4.2. ncRNA Biomarkers—Cohort Studies

Many have suggested that ncRNAs may also act as markers in predicting the development and recurrence of various cancers and patient response to therapy [119]. There has been a lot of work on the existence of these correlations, and one example is the analysis performed by Mitash et al. in 2017. It showed that there was a correlation between high miR-21 expression in tumour tissues and shortened time to recurrence-free survival (RFS) (HR = 0.64, 95% CI; log rank = 0.03). MiR-21 negatively affects the expression of *PTEN*, a tumour suppressor gene, contributing to the uncontrolled development of BC [139]. A 2016 paper by Zhang X. et al. showed that high miR-155 expression in urine samples of BC patients correlated with shortened RFS (HR = 3.497; 95% CI: 1.722–7.099; *p* = 0.001) and progression-free survival (PFS) (HR = 10.146; 95% CI: 389–74.091; *p* = 0.022). MiR-155 causes inhibition of *APC* gene expression, and this in turn affects Wnt/β-catenin pathway activation and BC progression [98]. Another analysis, which in turn observed the effect of low miR-133b expression in tissues from BC patients on shorter PFS (RR = 2.97; 95% CI: 1.78–6.44; *p* = 0.009) and OS (RR = 4.23; 95% CI: 1.51–11.8; *p* = 0.011), is the study by Chen X. et al. in 2016. MiR-133b activates the *EGFR* gene, which induces the mitogen-activated protein kinase/extracellular signal-regulated kinase (MAPK/ERK) and phosphoinositide-3-kinase/protein kinase B/Akt (PI3K/AKT) signalling pathways. The result is excessive proliferation of cancer cells [140,141]. A 2018 paper by Avgeris M. et al. reported that patients with reduced lncRNA-GAS5 expression in tumour-lesioned tissues showed significantly shorter PFS (HR = 6.628; 95% CI: 1.494–29.40; *p =* 0.013). Silencing of lncRNA-GAS5 contributed to an increased proportion of tumour cells in the S/G2 phase of the cell cycle, which was associated with overexpression of the *CDK6* gene, responsible for cycle regulation [142]. Jiao R. et al. in 2020 indicated that increased expression of lncRNA-SNHG16 in tested tumour tissues was significantly associated with decreased OS in patients (HR = 2.523, 95% CI: 1.540–4.133, *p* < 0.001). By interacting with *CCL5*, LncRNA-SNHG16 disrupts signalling pathways, affecting chemokine expression and creating a microenvironment that promotes BC growth [143]. Chen C. et al., in 2020, addressed a similar issue, observing a correlation between overexpression of lncRNA-LNMAT2 in tumour-transformed tissues and shorter OS (HR = 0.62; 95% CI: 0.42–0.91; *p* < 0.05). Overexpression of lncRNA-LNMAT2 activates the *PROX1* gene, which promotes BC progression and lymph node metastasis [144]. By contrast, Lin G. et al. in 2019 observed an association between low expression of circRNA-LPAR1 in tissues of BC patients and shortened disease-specific survival (DSS, HR = 0.364; 95% CI: 0.197–0.673; *p =* 0.001). Circ-LPAR1 interacting with the *WNT5A* gene activates the Wnt/β-catenin pathway and leads to BC cell proliferation [145,146]. A similar study was conducted by Tang G. et al. in 2017, where they evaluated the relationship between high expression of circRNA-ASXL1 in tumour tissue and reduced OS (HR = 4.831; 95% CI: 1.718–13.514; *p =* 0.007). CircRNA-ASXL1 affects the *TP53* gene by deregulating the mechanism of action of the p53 protein, resulting in disruption of the body’s defence against BC development [104].

Noteworthy is the fact that the topic of ncRNA research and BC prediction is constantly being expanded with new studies that provide a lot of valuable information. A paper by Andrew AS. et al. in 2019 showed that low expression of miR-26b-5p in tumour tissue was associated with reduced PFS in BC patients (HR = 0.043, 95% CI; 0.0079–0.24, *p =* 0.0008). MiR-26b-5p affects the level of the *PLOD2* gene, which in turn is involved in collagen maturation. In case of aberrant expression of *PLOD2*, stiffness and dysregulation of the extracellular matrix occur, and further migration and invasion of BC cells follows [147,148]. Borkowska et al. in 2019 also focused on demonstrating an association between high miR-21-5p expression in bladder cancer tissues and shortened overall survival (OS, HR = 1.00009, 95% CI: 1.000025–1.00015, *p =* 0.0069). MiR-21-5p may interact with the *TP53* gene to inhibit the mechanisms controlled by p53, reducing the effectiveness of the cell’s natural defences against tumorigenesis [149]. On the other hand, an analysis by Juracek J. et al. in 2019 showed a correlation between reduced miR-34a-3p expression in tissues and shortened OS in patients with BC (HR = 0.3184, 95% CI: 0.003–0.681, *p =* 0.0258). MiR-34a-3p interacts with the tumour suppressor gene *PTEN*, contributing to BC progression [150,151]. As another example, a study by Yang L. et al. in 2021 observed a correlation between high miR-10a-5p expression in the plasma of bladder cancer patients and reduced OS (HR = 1.74, 95% CI: 1.31–2.01, *p =* 0.002). MiR-10a-5p regulates the expression of the *FGFR3* gene, which, when activated, causes dimerisation of fibroblast growth factor receptor 3, which promotes BC cell proliferation and survival [149,152,153]. Yin et al. 2019 noted that elevated miR-185 expression in tumour tissues of BC patients was correlated with shorter OS (HR = 0.820; 95% CI: 0.688–0.978; *p =* 0.027). MiR-185, through interaction with the *ITGB5* gene, may lead to excessive proliferation and invasion of tumour-lesioned cells [154]. In 2022, Yerukala S. et al. also analysed the correlation between high miR-652-5p expression in tumour tissues and shortened OS (HR = 0.54, 95% CI: 0.40–0.72, *p =* 0.00004). The effect of miR-652-5p on the *KCNN3* gene, which regulates the flow of potassium ions in biological membranes, can lead to abnormal membrane potential, resulting in abnormal cell proliferation [155,156]. By contrast, Chen C. et al. in 2021 showed that overexpression of lncRNA-ELNAT1 in urine was correlated with reduced OS in patients with BC (HR = 1.76; 95% CI: 1.21–2.55; *p* < 0.01). LncRNA-ELNAT1, like lncRNA-LNMAT2, also promotes lymphatic metastasis by affecting the *UBC9* gene in bladder cancer [157]. Liang T. et al. in 2021 noted that there was a correlation between high expression of lncRNA-MALAT1a and shortened OS (HR = 3.278; 95% CI: 2.159–3.430; *p =* 0.004). LncRNA-MALAT1, through its activating effect on the *MDM2* gene, can lead to decreased p53 activity, which is associated with BC progression [158]. In a study by Zheng H. et al. in 2021, an association between lncRNA-BCYRN1 overexpression in tissues of BC patients and shorter OS was observed (HR = 1.58; 95% CI: 1.07–2.33; *p* < 0.05). This was because increased lncRNA-BCYRN1 expression targeted the *WNT5A* gene, contributed to activation of the Wnt/β-catenin signalling pathway, and consequently to the growth and migration of BC cells [159]. Another study, also related to proving a correlation between high lncRNA- TERC expression in BC patient tissues and reduced OS, is the analysis by Chen C. et al. in 2022 (HR = 1.51; 95% CI: 0.95–2.39). Overexpressed lncRNA-TERC interacts with the TERT gene, promoting increased telomerase activation in cancer cells and leading to uncontrolled BC progression [88,160]. A study by Xie F. et al. in 2020 reported that low expression of circRNA-HIPK3 in BC tissues was associated with shorter OS in these patients (HR = 0.4155; 95% CI: 0.2148–0.8036; *p =* 0.0091). Silencing of circRNA-HIPK3 contributed to the inhibition of *SOX4* gene expression, which regulates mesenchymal features and may result in an increased ability to migrate and metastasise in BC patients [161,162]. Analogous results were obtained by Su Y. et al. in 2020, as they showed that low circRNA-RIP2 expression in tumour-lesioned tissues was associated with shortened OS in BC patients (HR = 0.339; 95% CI: 0.128–0.898; *p =* 0.029). Reduced circRNA-RIP2 expression deregulates normal *SMAD3* gene expression, leading to uncontrolled activation of the TGF-β/SMAD signalling pathway, and this promotes BC cell invasion [163]. A similar relationship was studied by Zhu J. et al. in 2021 in the context of circRNA-EHBP1 and OS. As a result of the analyses, they showed that high expression of this molecule in tissues from BC patients was associated with shorter OS (HR = 0.48; 95% CI: 0.31–0.74; *p* < 0.01). CircEHBP1 increases *TGFβR1* gene expression, thereby activating the TGF-β/SMAD3 signalling pathway, which promotes lymph angiogenesis in BC [164].

A summary of the ncRNA-based biomarkers discussed and proposals for biomarkers not mentioned in the text are provided in Table 5 and Figure 2.

## 5. Strength and Limitations

One of the main strengths of this review is the wide range of research papers included in the analysis, as evidenced by the extensive bibliography, which focuses mainly on recent work, thus providing a solid update of the current state of knowledge. The selection of papers analysed was based on several different epigenetic mechanisms with high potential for use as diagnostic and prognostic biomarkers of BC, allowing a wide range of data to be presented. This review article summarises and analyses data on the diagnostic potential of selected biomarkers, which are presented in a clear and lucid manner and can be a valuable resource for those wishing to learn about the current state of knowledge and trends discussed in this field. On the other hand, we are aware of the limitations of this narrative review. Selection of the literature was based mainly on English language articles, which may exclude important studies available in other languages. In addition, the focus was on specific indicators for assessing the diagnostic and prognostic utility of epigenetic biomarker data, while others were discarded. Attention was focused primarily on recent articles, which may have excluded older studies that are relevant to the field. Due to these limitations, the findings of the review should be interpreted with caution.

## 6. Conclusions

Early diagnosis and accurate assessment of prognosis are key to successful treatment of BC. The research findings included in this review demonstrate the high potential of epigenetic alterations as diagnostic and prognostic biomarkers in BC. For early-stage cancers, DNA methylation-based biomarkers are of particular importance Recent scientific work indicates the high potential of hypermethylated genes *TWIST1*, *VIM*, *ZNF671*, *OTX1*, and *IRF8* for early diagnosis and *PTK2*, *AHNAK*, and *DAPK* for BC prognosis. On the other hand, among ncRNAs, a panel of three miRNA molecules, miR-20a-5p, miR-921-3p, and miR-17-5p, or lncRNA RMRP, among others, had the highest sensitivity and specificity values for detecting BC. The ncRNA molecules also appear to show potential in the prognosis of BC. Recently, the utility of circRNA CCT3, lncRNA-HOX, and miR-205-3P as promising prognostic biomarkers of BC has been proposed. In addition, some epigenetic changes, such as miR-155 and circRNA CCT3, are useful in both early detection and prognosis of BC. An important advantage of the epigenetic changes discussed is that they can be studied in readily available biological materials such as blood or urine. However, despite the promising results of these studies, epigenetic alterations are still not part of the standard methods for diagnosing and predicting BC, e.g., due to the study groups being too small, lack of reproducibility of results, lack of adequate validation of the results obtained, and the need for appropriate facilities, equipment, and highly qualified personnel, which can generate high costs. On the other hand, finding an epigenetic biomarker with high sensitivity and specificity could, in practice, be more cost efficient than a standardised expensive cystoscopy. In addition, it is worth noting that finding an effective biomarker to detect BC at an early stage of development could dramatically reduce the costs associated with both diagnosis and treatment of the patient itself. Given the above, further research, especially conducted on sufficiently large groups of patients, taking into account the validation of the results obtained, as well as aiming to optimise diagnostic methods and obtain biomarkers that combine all of the characteristics of a desirable BC biomarker, seems fully justified.

## Figures and Tables

**Figure 1 jcm-13-07159-f001:**
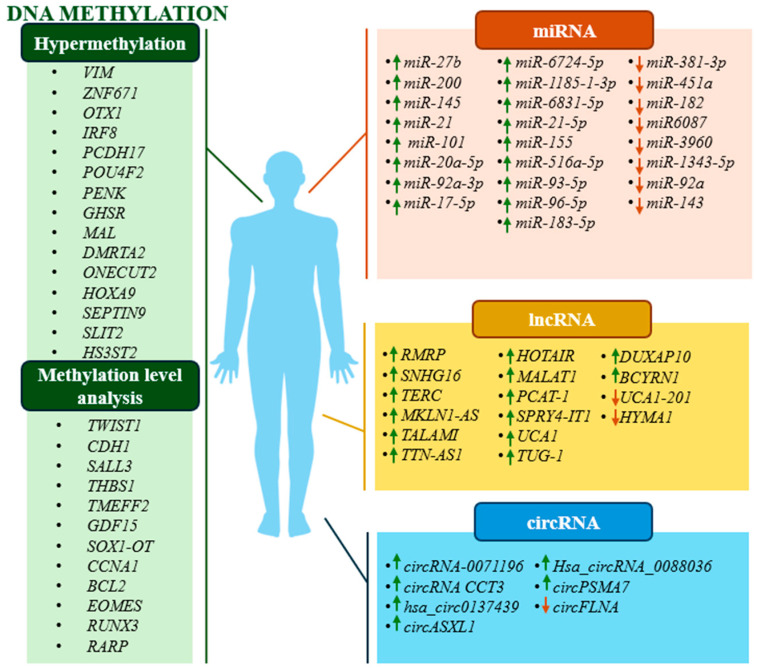
Schematic representation of studied changes in methylation status and expression levels of potential diagnostic biomarkers. Green arrows indicate an increase in the expression level and red arrows indicate a decrease in the expression level of a particular ncRNA.

**Figure 2 jcm-13-07159-f002:**
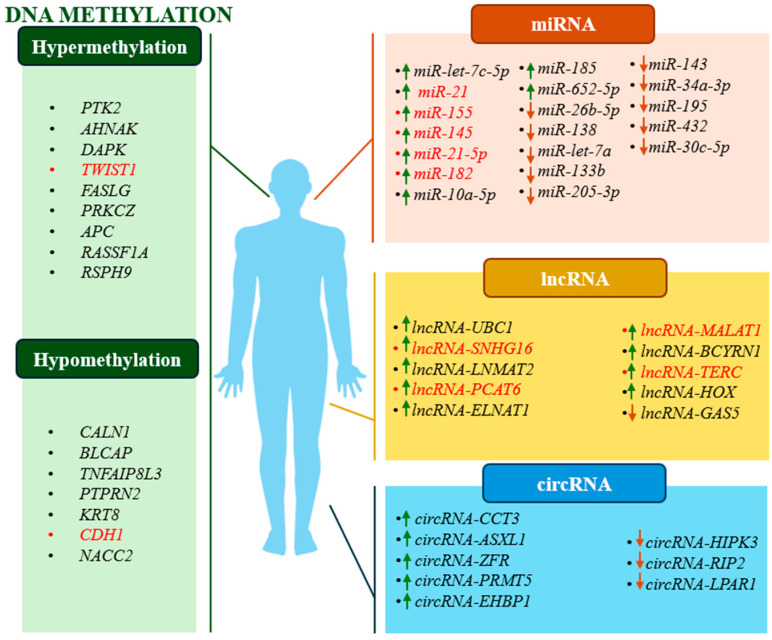
Schematic representation of studied changes in methylation status and type of regulation of potential prognostic biomarkers. Green arrows indicate upregulation and red arrows indicate downregulation of a given ncRNA. Red text indicates examples of biomarkers that show both diagnostic and prognostic potential.

**Table 1 jcm-13-07159-t001:** Available tests based on DNA methylation for BC diagnosis.

Authors	Test Name	Target Biomarker	Method	Study Material	Diagnostic Performance	Study Group (n)	Race/ Nationality
Bang et al. (2024) [56]	EarlyTest BCD	*PENK*	qPCR	Voided urine	SN = 81% SP = 91.5%	210 (21 BC, 189 non-BC based on cystoscopy)	Korean, American
Pharo et al. (2022) [54]	BladMetrix	8 methylated gene regions	ddPCR	Urinary exfoliated cell DNA	SN = 92.1% SP = 93.3%	273 (with gross haematuria) (93 BC)	European
Piatti et al. (2021) [55]	Bladder CARE	DNA methylation in *TRNA-Cys*, *SIM2*, and *NKX1*	Methylation-sensitive restriction enzymes with qPCR	Urine	SN = 93.5% SP = 92.6%	213 (77 BC, 136 non-BC)	Caucasian, Asian, African American, Hispanic
Steinbach et al. (2020) [57]	GynTect	DNA methylation in *ASTN1*, *DLX1*, *ITGA4*, *RXFP3*, *SOX17*, and *ZNF671*	GynTect Assay	Urine	SN = 60% SP = 96.7%	70 (40 BC, 15 BPH, 15 Urolithiasis)	German
Witjes et al. (2018) [52]	Bladder EpiCheck	15 methylated gene regions	qPCR	Urinary exfoliated cell DNA	SN = 68.2% SP = 88%	353 (UC)	Caucasian
Feber et al. (2017) [53]	UroMark	150 CpG loci methylation	Targeted bisulfite sequencing	Urinary exfoliated cell DNA	SN = 98% SP = 97%	274 (107 BC, 167 non-BC)	English

qPCR—quantitative polymerase chain reaction, ddPCR—droplet digital PCR, SN—sensitivity, SP—specificity, BC—bladder cancer, BPH—benign prostatic hyperplasia, UC—urothelial carcinoma.

**Table 2 jcm-13-07159-t002:** DNA methylation-based biomarkers for BC diagnosis validated in cohort studies.

Authors	Study Biomarker	Methylation Status	Study Material	Diagnostic Performance	Study Group (n)	Race/ Nationality
Zhang et al. (2024) [64]	*TWIST1*, *VIM*	Methylation level analysis	Urine cell sediment	SN = 78% SP = 83%	231 (77 BC, 81 other urological malignancies, 19 benign disease, 26 UTUC, 28 healthy)	China
Jiang et al. (2024) [61]	*ZNF671*, *OTX1*, *IRF8*	Hypermethylation	Voided urine	SN = 75% SP = 90.9%	114 (61 BC, 53 non-BC)	Taiwan
Fang et al. (2022) [59]	*PCDH17*, *POU4F2*, *PENK*	Hypermethylation	Urine cell sediment	SN = 87% SP = 97%	207 (107 BC, 100 non-BC)	China
Hentschel et al. (2022) [60]	*GHSR*, *MAL*	Hypermethylation	Urine pellet	SN = 80% SP = 93%	208 (108 BC, 34 benign haematuria, 43 other benign urological conditions, 23 healthy)	Dutch
Deng et al. (2022) [65]	*DMRTA2*	Hypermethylation	Urine	SN = 82.9% SP = 92.5%	520 (79 BC, 107 other malignancies, 22 benign tumours of bladder8 recurring cancers, 304 healthy	China
Ruan et al. (2021) [66]	*ONECUT2*, *VIM*	Hypermethylation	Urine	Cohort 1: SN = 88.1% SP = 89.7% Cohort 2: SN = 91.2% SP = 85.7%	Cohort 1: 98 (patients suspected of BC) (59 BC, 39 non-BC); Cohort 2: 174 (haematuria patients) (34 BC, 140 non-BC)	China
Wu et al. (2020) [62]	*ONECUT2*, *HOXA9*, *PCDH17*, *POU4F2*	Hypermethylation	Urine	SN = 90.5% SP = 73.2%	111 (53 BC, 58 non-BC)	China
Chen et al. (2020) [63]	*OTX1*, *SOX1-OT*	Methylation level analysis	Urine	SN = 91.7% SP = 77.3%	175 (109 BC, 66 benign diseases)	China
Guo et al. (2018) [67]	*VIM*, *CDH1*, *SALL3*, *THBS1*, *TMEFF2*, *GDF15*	Methylation level analysis	Voided urine	SN = 89% SP = 74%	473 (217 UC, 256 controls)	China
Roperch et al. (2016) [68]	*SEPTIN9*, *SLIT2*	Hypermethylation	Urine	SN = 91% SP = 71.4%	272 (167 NMIBC, 105 controls)	France
	*HS3ST2*, *SEPTIN9*, *SLIT2*	Hypermethylation	Urine	SN = 90.4% SP = 75.2%	272 (167 NMIBC, 105 controls)	France
	*HS3ST2*, *SEPTIN9*	Hypermethylation	Urine	SN = 90.4% SP = 72.4%	272 (167 NMIBC, 105 controls)	France
Dahmcke et al. (2016) [69]	*CCNA1*, *ONECUT2*, *BCL2*, *EOMES*, *SALL3*, *VIM*	Methylation level analysis	Urine	SN = 89.9% SP = 88.6%	475 (99 BC, 376 controls)	Denmark
Wang et al. (2016) [70]	*RUNX3*, *RARP*	Methylation level analysis	Urine	SN = 96.6% SP = 88.9%	139 (112 BC, 10 healthy, 17 glandular cystitis)	China

SN—sensitivity, SP—specificity, BC—bladder cancer, UC—urothelial carcinoma, UTUC—upper tract urothelial carcinoma.

**Table 3 jcm-13-07159-t003:** NcRNA-based biomarkers for BC diagnosis validated in cohort studies.

Authors	Study Biomarker	Biomarker Change	Biomarker Targets	Study Material	Diagnostic Performance	Study group (n)	Race/ Nationality
**miRNAs**							
Lu et al. (2024) [82]	miR-221-5p, miR-181a-5p, miR-15a-5p, miR-222-3p	Aberrantly expressed	Not specified	Serum	SN = 82.1% SP = 85.7%	224 (112 BC, 112 controls)	China
Yu et al. (2023) [73]	miR-27b, miR-381-3p, miR-451a	Overexpression: miR-27b Underexpression: miR-381-3p, miR-451a	*SMAD4*, *FOXO1*	Serum	SN = 86.7% SP = 77.4%	224 (112 BC, 112 healthy)	China
Mamdouh et al. (2023) [91] * Zhang et al. (2018) [92] ** Zhang et al. (2015) [93]	miR-200	Overexpression	Not specified	Urine and tissue	Tissue: SN = 93.3% SP = 100% Urine: SN = 62.2% SP = 100%	136 (111 BC, 25 healthy)	Egyptian
	miR-145		N-cadherin *		Tissue: SN = 80% SP = 100% Urine: SN = 78.4% SP = 91.7%		
	miR-21		maspin, *VEGF-C ***		Tissue: SN = 73.3% SP = 80% Urine: SN = 83.3% SP = 100%		
SurezCabrera et al. (2022) [74]	miR-145, miR-182	Overexpression (miR-145), Underexpression (miR-182)	*FCS1*	Urine	SN = 93% SP = 86%	82 (40 BC, 42 controls)	European
Wang et al. (2021) [71]	miR-20a-5p, miR-92a-3p, miR-17-5p	Overexpression	p21, *PTEN*	Serum	SN = 90.4% SP = 94.4%	164 (74BC, 90 healthy)	China
El-Shal et al. (2021) [75]	miR-96-5p, miR-183-5p	Overexpression	*FOXO*, *KRAS PDCD4*	Voided urine	miR-96 alone: SN = 80.4% SP = 91.8% miR-183 alone: SN = 78.4% SP = 81.6% both combined: SN = 88.2% SP = 87.8%	100 (51 BC, 21 benign bladder lesions, 28 healthy)	Egyptian
Lin et al. (2021) [79]	miR-516a-5p, miR-93-5p	Overexpression	miR-516a-5p (not specified) miR-93-5p: *PEDF*, *EGFR*, FoxO pathway, PI3K-Akt pathway, *BTG2*	Midstream urine	miR-93-5p alone: SN = 74.1% SP = 90.2% miR-516a-5p alone: SN = 72.9% SP = 89.9% both combined: SN = 85.2% SP = 82.4%	104 (53 BC, 51 healthy)	China
Piao et al. (2019) [94]	miR-6124, miR-4511	Aberrantly expressed	Not specified	urine	SN = 91.5% SP = 76.2% (ratio miR-6124 to miR-4511)	543 (326 BC, 174 haematuria, non-BC pyuria)	Republic of Korea
Chen (2019) [81]	miR-101	Underexpression	Not specified	serum	SN = 82% SP = 80.9%	232 (122 BC, 110 healthy)	China
Usuba et al. (2018) [95]	miR-6087, miR-6724-5p, miR-3960, miR-1343-5p, miR-1185-1-3p miR-6831-5p, miR-4695-5p combined	miR-4695-5p: no significant change Underexpression: miR6087, miR-3960, miR-1343-5p Overexpression: miR-6724-5p, miR-1185-1-3p, miR-6831-5p	Not specified	serum	SN = 95% SP = 87%	972 (392 BC, 100 non-BC, 480 other cancers)	Japan
Huang et al. (2018) [96]	miR-20a	Overexpression	Not specified	Urine	SN = 72.1% SP = 87.5%	166 (80 NMIBC, 86 healthy)	China
Matsuzaki et al. (2017) [80]	miR-21-5p	Overexpression	Not specified	Urine	SN = 75% SP = 98%	60 (24 controls, 36 UC)	Japan
Urquidi et al. (2016) [97]	25 miRNAs combined	Aberrantly expressed	Not specified	Midstream urine	SN = 87% SP = 100%	121 (61 cases, 60 controls)	USA
Zhang et al. (2016) [98]	miR-155	Overexpression	*APC*, *VHL*, *PIK3R1*, *MLH1*	Voided urine	SN = 80.2% SP = 84.6%	324 (162 NIMBC, 86 cystitis, 76 healthy)	China
Motawi et al. (2016) [99]	miR-143, miR-92a	Underexpression	Not specified	Plasma	SN = 94.3%, SP = 86.6%	132 (70BC, 62 healthy)	Egyptian
**lncRNAs**							
Gao et al. (2024) [87]	RMRP	Overexpression	Not specified	Urine	SN = 83% SP = 70% (RT-qPCR) SN = 95% SP = 92.5% (RT-RAA- CRISPR/Cas12a)	339 (229 BC, 110 benign lesions)	China
Liu et al. (2023) [100]	SNHG16	Overexpression	possibly Wnt/β-catenin pathway	Urine	SN = 61.9% SP = 83.3%	84 (42 BC, 42 Healthy)	China
Chen et al. (2022) [88]	TERC	Overexpression	Not specified	Urine	SN = 78.7% SP = 77.8%	152 (89 BLCA, 63 Healthy)	China
Bian et al. (2022) [101]	MKLN1-AS	Overexpression	Not specified	Urine	SN = 79.1% SP = 67.4%	92 (46 BC, 46 Controls)	China
	TALAM1		Not specified		SN = 90.1% SP = 55.8%		
	TTN-AS1		Not specified		SN = 76.7% SP = 76.7%		
	UCA1		PI3K-Akt-mTOR pathway, *GLS2*, *HMGB1*, p21		SN = 90.7% SP = 90.7%		
Sarfi et al. (2021) [102]	TUG-1	Overexpression	Not specified	Voided urine	SN = 76.7% SP = 77.8%	40 (30 NMIBC, 10 Controls)	Iran
Yu et al. (2020) [86]	UCA-1-201, HOTAIR, HYMA1, MALAT1	Overexpression: HOTAIR, MALAT1 Underexpression: UCA-1-201, HYMA1	Not specified	Urine	SN = 93.3% SP = 96.7%	120 (60 Urocystitis, 60 NMIBC)	China
Zhan et al. (2018) [85]	MALAT1, PCAT-1, SPRY4-IT1	Overexpression	Not specified	Urine	SN = 62.5% SP = 85.6%	208 (104 BC, 104 healthy)	China
**other ncRNAs**							
Yang et al. (2024) [89]	circRNA-0071196	Overexpression	*CIT*, miR-19b-3p,	Urine	SN = 87.5% SP = 85%	70 (40 BUC, 30 non-BUC)	China
Luo et al. (2023) [103]	circRNA CCT3	Overexpression	*PP2A*, miR-135a-5p	Plasma	SN = 86.1% SP = 60%	125 (85 BC, 40 healthy)	China
Song et al. (2020) [90]	hsa_circ0137439	Overexpression	miR-142-5p	Urine	SN = 87.9 SP = 80.1% (BC vs. Controls) SN = 88.6% SP = 73.5% (NMIBC vs. MIBC)	146 (62 NMIBC, 54 MIBC, 30 controls)	China
Tang et al. (2017) [104]	circASXL1	Overexpression	Not specified	Tumour tissue	SN = 68.6% SP = 76.9%	61 pairs of tumour tissue and adjacent normal mucosa	China

SN—sensitivity, SP—specificity, BC—bladder cancer, BUC bladder urothelial carcinoma, NMIBC—non-muscle-invasive bladder cancer, MIBC—muscle-invasive bladder cancer, UC—urothelial carcinoma, *—information with reference to Zhang et al. (2018), **—information with reference to Zhang et al. (2015).

**Table 4 jcm-13-07159-t004:** DNA methylation-based biomarkers for BC prognosis validated in cohort studies.

Authors	Study Biomarker	Methylation Status	Study Material	Assessed Study Endpoints (Associated Change)	Study Group (n)	Race/ Nationality
Kim et al. (2024) [133]	*PTK2*	hypermethylation	Tissue	OS (↓)	BC patients (n = 275) healthy donors (n = 10)	Republic of Korea
Zhang et al. (2024) [129]	*AHNAK*	hypermethylation	Tissue	OS (↓)	BC patients (n = 812)	China
Koukourikis et al. (2023) [135]	*DAPK*	hypermethylation	Urine	OS (↓)	BC patients (n = 414) healthy donors (n = 10)	Greece
El Azzouzi et al. (2022) [136]	*TWIST1*	hypermethylation	Tissue	PFS (↓)	BC patients (n = 70)	Morocco
Takagi et al. (2022) [130]	*CALN1*	hypomethylation	Tissue	PFS (↓)	BC patients (n = 82)	Japan
Zhang et al. (2022) [134]	*FASLG*, *PRKCZ*	hypermethylation	Tissue	PFS (↓)	BC patients (n = 408) healthy donors (n = 14)	China
Chen et al. (2022) [131]	*BLCAP*	hypomethylation	Peripheral blood	PFS (↓) OS (↓)	BC patients (n = 603)	USA
Guo et al. (2021) [137]	*TNFAIP8L3*	hypomethylation	Tissue	PFS (↓) OS (↓)	BC patients (n = 357)	China
Zhou et al. (2021) [138]	*PTPRN2*	hypomethylation	Tissue	OS (↓)	BC patients (n = 399)	China
Guo et al. (2021) [137]	*APC*	hypermethylation	Tissue	PFS (↓)	BC patients (n = 357)	China
Zhang et al. (2020) [128]	*KRT8*	hypomethylation	Tissue	OS (↓)	BC patients (n = 41) healthy donors (n = 35)	China
Zhang et al. (2018) [128]	*CDH1*	hypomethylation	Tissue	OS (↓)	BC patients (n = 167) healthy donors (n = 13)	China
Zhan et al. (2017) [126]	*RASSF1A*	hypermethylation	Tissue	PFS (↓) OS (↓)	BC patients (n = 389)	China
Shivakumar et al. (2017) [127]	*NACC2*	hypomethylation	Tissue	OS (↓)	BC patients (n = 403)	USA
Yoon et al. (2016) [125]	*RSPH9*	hypermethylation	Tissue	PFS (↓)	BC patients (n = 128) healthy donors (n = 8)	Republic of Korea

BC—bladder cancer, OS—overall survival, PFS—progression-free survival, ↓—decrease of Assessed Study Endpoints.

**Table 5 jcm-13-07159-t005:** The ncRNA-based biomarkers for BC prognosis validated in cohort studies.

Authors	Study Biomarker	Biomarker Change	Biomarker Targets	Study Material	Assessed Study Endpoints (Associated Change)	Study Group (n)	Race/ Nationality
**miRNAs**							
Zhenhai et al. (2023) [165]	miR-205-3p	Downregulated	*GLO1*	Tissue	PFS (↓) OS (↓)	BC patients (n = 35)	China
Hao et al. (2023) [166]	miR-30c-5p	Downregulated	*PRC1*	Tissue	OS (↓)	BC patients (n = 445)	China
Awadalla et al. (2022) [167]	miR-138	Downregulated	*HIF1α*	Tissue	CSS (↓)	BC patients (n = 157)	Egypt
Awadalla et al. (2022) [167]	miR-let-7a	Downregulated	*WNT7A*	Tissue	CSS (↓)	BC patients (n = 157)	Egypt
Yerukala et al. (2022) [155]	miR-652-5p	Upregulated	*KCNN3*	Tissue	OS (↓)	BC patients (n = 106)	USA
Zhang et al. (2021) [168]	miR-432	Downregulated	*SMARCA5*	Tissue	OS (↓)	BC patients (n = 156)	China
Yang et al. (2021) [152] Borkowska et al. (2019) [149]	miR-10a-5p	Upregulated	*FGFR3*	Plasma	OS (↓)	BC patients (n = 88) healthy donors (n = 36)	China
Andrew et al. (2019) [147]	miR-26b-5p	Downregulated	*PLOD2*	Tissue	PFS (↓)	BC patients (n = 231)	Lebanon
Spagnuolo et al. (2020) [169]	miR-let-7c-5p	Upregulated	*HRAS*	Urine	PFS (↓)	BC patients (n = 57) healthy donors (n = 20)	Italy
Borkowska et al. (2019) [149]	miR-21-5p	Upregulated	*TP53*	Tissue	OS (↓)	BC patients (n = 55) healthy donors (n = 30)	Poland
Braicu et al. (2019) [170]	miR-143	Downregulated	*TP53*	Tissue	OS (↓)	BC patients (n = 409) healthy donors (n = 19)	Romania
Chen et al. (2019) [171]	miR-182	Upregulated	*FOXO3a*	Tissue	OS (↓)	BC patients (n = 60) healthy donors (n = 20)	China
Juracek et al. (2019) [150] Ding et al. (2019) [151]	miR-34a-3p	Downregulated	*PTEN*	Tissue	OS (↓)	BC patients (n = 78)	Czech Republic
Yin et al. (2019) [154]	miR-185	Upregulated	*ITGB5*	Tissue	OS (↓)	BC patients (n = 408) healthy donors (n = 19)	China
Yang et al. (2019) [172]	miR-195	Downregulated	*MEK1*	Tissue	OS (↓)	BC patients (n = 80) healthy donors (n = 30)	China
Li et al. (2018) [173]	miR-145	Upregulated	*CDK4*	Tissue	OS (↓)	BC patients (n = 127)	China
Mitash et al. (2017) [139]	miR-21	Upregulated	*PTEN*	Tissue	RFS (↓)	BC patients (n = 31)	India
Zhang et al. (2016) [98]	miR-155	Upregulated	*APC*	Urine	RFS (↓) PFS (↓)	BC patients (n = 162) healthy donors (n = 86)	China
Chen et al. (2016) [140]	miR-133b	Downregulated	*EGFR*	Tissue	PFS (↓) OS (↓)	BC patients (n = 146)	China
**lncRNAs**							
Martins et al. (2024) [174] Novikova et al. (2021) [175]	lncRNA-HOX	Upregulated	*HOXD*	Peripheral blood	OS (↓)	BC patients (n = 106) healthy donors (n = 199)	Portugal
Chen et al. (2022) [88]	lncRNA-TERC	Upregulated	*TERT*	Tissue	OS (↓)	BC patients (n = 89) healthy donors (n = 63)	China
Chen et al. (2021) [157]	lncRNA-ELNAT1	Upregulated	*UBC9*	Urine	OS (↓)	BC patients (n = 242)	China
Liang et al. (2021) [158]	lncRNA-MALAT1	Upregulated	*MDM2*	Tissue	OS (↓)	BC patients (n = 90)	China
Zheng et al. (2021) [159]	lncRNA-BCYRN1	Upregulated	*WNT5A*	Tissue	OS (↓)	BC patients (n = 210)	China
Jiao et al. (2020) [143]	lncRNA-SNHG16	Upregulated	*CCL5*	Tissue	OS (↓)	BC patients (n = 1148)	China
Chen et al. (2020) [144]	lncRNA-LNMAT2	Upregulated	*PROX1*	Tissue	OS (↓)	BC patients (n = 266)	China
Xia et al. (2020) [176]	lncRNA-PCAT6	Upregulated	*MIR513A1*	Tissue	OS (↓)	BC patients (n = 21) healthy donors (n = 21)	China
Zhang et al. (2019) [177]	lncRNA-UBC1	Upregulated	*PRC2*	Serum	RFS (↓)	BC patients (n = 260) healthy donors (n = 260)	China
Avgeris et al. (2018) [142]	lncRNA-GAS5	Downregulated	*CDK6*	Tissue	PFS (↓)	BC patients (n = 176)	Greece
**other ncRNAs**							
Luo et al. (2023) [103]	circRNA-CCT3	Upregulated	*PP2A*	Urine	RFS (↓) OS (↓)	BC patients (n = 85) healthy donors (n = 40)	China
Luo et al. (2021) [178]	circRNA-ZFR	Upregulated	*WNT5A*	Tissue	OS (↓)	BC patients (n = 60)	China
Chen et al. (2021) [179]	circRNA-PRMT5	Upregulated	*SNAIL1*	Tissue	OS (↓)	BC patients (n = 119)	China
Zhu et al. (2021) [164]	circRNA-EHBP1	Upregulated	*TGFβR1*	Tissue	OS (↓)	BC patients (n = 186)	China
Xie et al. (2020) [161] Zhang et al. (2019) [162]	circRNA-HIPK3	Downregulated	*SOX4*	Tissue	OS (↓)	BC patients (n = 68)	China
Su et al. (2020) [163]	circRNA-RIP2	Downregulated	*SMAD3*	Tissue	OS (↓)	BC patients (n = 58)	China
Lin et al. (2019) [145]	circRNA-LPAR1	Downregulated	*WNT5A*	Tissue	DSS (↓)	BC patients (n = 125)	China
Tang et al. (2017) [104]	circRNA-ASXL1	Upregulated	*TP53*	Tissue	OS (↓)	BC patients (n = 61)	China

BC—bladder cancer, OS—overall survival, PFS—progression-free survival, RFS—recurrence-free survival, CSS—cause-specific survival, DSS—disease-specific survival, ↓—decrease of Assessed Study Endpoints.

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
