# Peer review of "Epigenetic Biomarkers as a New Diagnostic Tool in Bladder Cancer—From Early Detection to Prognosis"

_jcm, 2024, doi:10.3390/jcm13237159_

Round 1

Reviewer 1 Report

Comments and Suggestions for Authors

Dear Authors,

I read with interest your paper. However, I think that some points should be expanded

- materials and methods section: the review is interesting, but we need to clearly list how it was conducted, which keywords were used and where articles were searched. Then, how the articles were selected? Is it a narrative or systematic review? A summary or a section regarding these aspects should be present in the manuscript

- Strength and limitations: with the absence of the methodology, the article cannot be evaluated well. Together with the methodology, you should report limitations and how eventually to improve the search or to justify why some articles were discarded

Author Response

Comments 1: materials and methods section: the review is interesting, but we need to clearly list how it was conducted, which keywords were used and where articles were searched. Then, how the articles were selected? Is it a narrative or systematic review? A summary or a section regarding these aspects should be present in the manuscript

Response 1: Thank you very much for pointing this out. We fully agree that for a better understanding of how the literature selection was carried out, a section briefly describing the methodology for carrying out the review is essential. We have therefore added a section describing the methodology located on page 5, paragraph 2, line: 194-224: “This paper is based on a literature review of epigenetic biomarkers for bladder cancer focusing on the potential usefulness of selected alterations for diagnosis and prognosis in BC. This narrative review focuses primarily on data on epigenetic alterations such as methylation levels of specific genes or combinations of genes and expression levels of ncRNA molecules such as micro-RNA, lnc-RNA and circ-RNA. Articles were searched through databases such as “Pubmed” and “Google Scholar” in order to access scientific papers related to the topics covered. The search for relevant publications was based on several keywords and their combinations. The main keywords used in the search included “bladder cancer”, “biomarkers”, “epigenetics”, “early detection”, “prognosis” and “epigenetic biomarkers”. The selection of relevant articles focused on works published in the last eight years. The articles included in the bibliography, published earlier than 2016, were used for the introductory section, where the current state of knowledge on the epidemiology, etiology, treatment, prognosis and issues concerning the classical approach to the diagnosis of BC were described. The main goal was to collect the latest discoveries in the field of the subject under discussion and update the state of knowledge, and therefore the focus was particularly on results published in recent years. In order to perform this review, we were looking for papers that included an assessment of diagnostic usefulness of the epigenetic biomarkers or their combinations. As for biomarkers used for BC detection, indicators such as sensitivity and specificity were chosen (with only biomarkers with sensitivity and specificity of no less than 60% being considered). As for biomarkers with potential for prognostic use, we chose articles with prognostic indicators such as OS (Overall Survival), PFS (Progression-Free Survival), CSS (Cancer-Specific Survival), DSS (Disease-Specific Survival) or RFS (Recurrance-Free Survival). Publications that did not include the aforementioned data of diagnostic or prognostic utility were not included in this review. Selected scientific papers were analyzed, and the data contained in them were grouped to clearly and transparently present potential epigenetic biomarkers of BC and their diagnostic and/or prognostic role. Through such analysis it was possible to draw conclusions about the latest trends and most promising epigenetic changes that could be used as novel diagnostic and prognostic tools in BC patients”.

Additionally, in order to align the information in the article with the methodology presented, information relating to cohort studies on the diagnostic potential of the epigenetic changes in question published in years earlier than 2016 has been removed from the text. In order to expand the knowledge presented so far in the article, the removed biomarkers have been replaced by other, more up-to-date ones. Accordingly, changes have been made to Table 2 (page 8 and 9, paragraph 3.2)  and Table 3 (page 11-14, paragraph 3.2) in the article. Changes relating to this issue have also been made in page 10, paragraph 3.2, line: 395-397 on page 10, while some of the text in paragraph 3.2, line: 397-401 on page 10 has been removed. Additionally in order to the changes we have made, we had to revise correct information contained in the Figure 1 (page 15, paraghraph 3.2). Bibliographic positions relating to deleted data, have been replaced by positions relating to new data, included in the tables 2 and 3. The revised bibliographic items are: 69, 70 (page 29, lines: 914-922), 81, 99 (page 30, lines: 944-946, 983-986). 

Comments 2: Strength and limitations: with the absence of the methodology, the article cannot be evaluated well. Together with the methodology, you should report limitations and how eventually to improve the search or to justify why some articles were discarded

Response 2: Thank you for this comment. According to comment two, we have added a short section discussing the strengths and limitations of our review work located on page 26, paragraph 6, line: 728-743: “One of the main strengths of this review is the wide range of research papers included in the analysis, as evidenced by the extensive bibliography, which focuses mainly on recent work, thus providing a solid update of the current state of knowledge. The selection of papers analyzed was based on several different epigenetic mechanisms with high potential for use as diagnostic and prognostic biomarkers of BC, allowing a wide range of data to be presented. This review article summarizes and analyzes data on the diagnostic potential of selected biomarkers, which are presented in a clear and lucid manner and can be a valuable resource for those wishing to learn about the current state of knowledge and trends discussed in this field. On the other hand, we are aware of the limitations of this narrative review. The selection of literature was based mainly on English-language articles, which may exclude important studies available in other languages. In addition, the focus was on specific indicators for assessing the diagnostic and prognostic utility of epigenetic biomarker data, while others were discarded. Attention was focused primarily on recent articles, which may have excluded older studies that are relevant to the field. Due to these limitations, the review should be interpreted with caution”.

All changes made are visible in the file ”jcm3285523_tracking changes”.

Reviewer 2 Report

Comments and Suggestions for Authors

The paper under evaluation has a very interesting topic: improvement of bladder cancer diagnosis and evaluation of prognosis by using epigenetic biomarkers.

Being a review of the literature, it is important for the readers to understand how the articles to be included in it were selected. Please describe shortly the methodology of selecting the papers to be evaluated and included in the present review.

The authors include a presentations of epidemiology, etiology, treatment and prognosis of bladder cancer, as well as a description of diagnosis of bladder cancer, both in the classic approach and using epigenetic biomarkers. However, urinary markers that are approved by regulatory institutions and commercially available such as NMP22, BTA or FISH are not mentioned in the review. These are innovative methods trying to improve bladder cancer diagnosis and diminish the  invasiveness of this process. Please revise.

Some paragraphs lack conciseness: e.g. lines 204-207: "Urine sediment cytology is also frequently used as a complementary test, demonstrating a sensitivity and specificity of 48% (16% for low-grade tumors and 84% for high-grade tumors) and 86%, respectively. However, despite a fairly high sensitivity for high-grade tumors, the sensitivity for low-grade tumors is only 16% [47].” Please revise the text and try combining some phrases or paragraphs in order to obtain a more concise flow.

Some comments regarding the financial implications of using epigenetic biomarkers and other innovative techniques should be included.

Author Response

Comments 1: Being a review of the literature, it is important for the readers to understand how the articles to be included in it were selected. Please describe shortly the methodology of selecting the papers to be evaluated and included in the present review.

Response 1: Thank you very much for pointing this out. We fully agree that for a better understanding of how the literature selection was carried out, a section briefly describing the methodology for carrying out the review is essential. We have therefore added a section describing the methodology located on page 5, paragraph 2, line: 194-224: “This paper is based on a literature review of epigenetic biomarkers for bladder cancer focusing on the potential usefulness of selected alterations for diagnosis and prognosis in BC. This narrative review focuses primarily on data on epigenetic alterations such as methylation levels of specific genes or combinations of genes and expression levels of ncRNA molecules such as micro-RNA, lnc-RNA and circ-RNA. Articles were searched through databases such as “Pubmed” and “Google Scholar” in order to access scientific papers related to the topics covered. The search for relevant publications was based on several keywords and their combinations. The main keywords used in the search included “bladder cancer”, “biomarkers”, “epigenetics”, “early detection”, “prognosis” and “epigenetic biomarkers”. The selection of relevant articles focused on works published in the last eight years. The articles included in the bibliography, published earlier than 2016, were used for the introductory section, where the current state of knowledge on the epidemiology, etiology, treatment, prognosis and issues concerning the classical approach to the diagnosis of BC were described. The main goal was to collect the latest discoveries in the field of the subject under discussion and update the state of knowledge, and therefore the focus was particularly on results published in recent years. In order to perform this review, we were looking for papers that included an assessment of diagnostic usefulness of the epigenetic biomarkers or their combinations. As for biomarkers used for BC detection, indicators such as sensitivity and specificity were chosen (with only biomarkers with sensitivity and specificity of no less than 60% being considered). As for biomarkers with potential for prognostic use, we chose articles with prognostic indicators such as OS (Overall Survival), PFS (Progression-Free Survival), CSS (Cancer-Specific Survival), DSS (Disease-Specific Survival) or RFS (Recurrance-Free Survival). Publications that did not include the aforementioned data of diagnostic or prognostic utility were not included in this review. Selected scientific papers were analyzed, and the data contained in them were grouped to clearly and transparently present potential epigenetic biomarkers of BC and their diagnostic and/or prognostic role. Through such analysis it was possible to draw conclusions about the latest trends and most promising epigenetic changes that could be used as novel diagnostic and prognostic tools in BC patients”.

Additionally, in order to align the information in the article with the methodology presented, information relating to cohort studies on the diagnostic potential of the epigenetic changes in question published in years earlier than 2016 has been removed from the text. In order to expand the knowledge presented so far in the article, the removed biomarkers have been replaced by other, more up-to-date ones. Accordingly, changes have been made to Table 2 (page 8 and 9, paragraph 3.2)  and Table 3 (page 11-14, paragraph 3.2) in the article. Changes relating to this issue have also been made in page 10, paragraph 3.2, line: 395-397 on page 10, while some of the text in paragraph 3.2, line: 397-401 on page 10 has been removed. Additionally in order to the changes we have made, we had to revise correct information contained in the Figure 1 (page 15, paraghraph 3.2). Bibliographic positions relating to deleted data, have been replaced by positions relating to new data, included in the tables 2 and 3. The revised bibliographic items are: 69, 70 (page 29, lines: 914-922), 81, 99 (page 30, lines: 944-946, 983-986). 

Comments 2: The authors include a presentations of epidemiology, etiology, treatment and prognosis of bladder cancer, as well as a description of diagnosis of bladder cancer, both in the classic approach and using epigenetic biomarkers. However, urinary markers that are approved by regulatory institutions and commercially available such as NMP22, BTA or FISH are not mentioned in the review. These are innovative methods trying to improve bladder cancer diagnosis and diminish the  invasiveness of this process. Please revise.

Response 2: Thank you for this comment. However, we would like to point out that our review was intended to focus only on selected epigenetic biomarkers of bladder cancer. Therefore, protein biomarkers or FISH technology based on the detection of chromosomal abnormalities were intentionally omitted from our article. However, with regard to the comment, we have added a paragraph to complement the reviewer's proposed elements, found on pages 7 and 8, paragraph 3.1, lines: 310-329: “Several diagnostic tests approved by the Food and Drug Administration (FDA) based on urine samples are currently available, but none of them are based on the epigenetic changes discussed in the article. Among them are four tests based on protein biomarkers and a test based on the examination of exfoliated cells. Protein biomarkers include BTA (Bladder Tumor Antigen) on the basis of which two diagnostic tests have been constructed. Qualitative BTA-Stat is a rapid immunochromatographic test (sensitivity and specificity of 64% and 77%, respectively) and BTA-TRAK is an ELISA immunoenzymatic test (sensitivity and specificity, 65% and 74%, respectively). Both tests detect and measure the level of human complement factor H-related protein (hCFHrp) in urine samples. Another FDA-approved BC protein biomarker is NMP22 (Nuclear Matrix Protein 22) on the basis of which two diagnostic tests have also been developed: the quantitative NMP22 ELISA (sensitivity and specificity, 69% and 77%, respectively) and the qualitative NMP22 BladderChek test (sensitivity and specificity, 58% and 88%, respectively), which have been validated for initial diagnosis. Also commercially available is the UroVysion test, which uses a multi-probe FISH (fluorescence in situ hybridization) technique to detect four chromosomal abnormalities that have been linked to the development of BC: aneuploidy of chromosomes 3, 7 and 7, and loss of the 9p21 locus. UroVysion achieves specificity UroVysion achieves a specificity of 87.7% and a sensitivity of 63%. Although the aforementioned tests have been approved by the FDA due to their limitations, they are not widely used in clinical practice [48, 59]”. We believe that supplementing the article with commercially available and approved by the regulatory institutions biomarkers may provide a broader view of the current state of knowledge and trends in the field of biomarkers used for bladder cancer diagnosis.

Comennts 3: Some paragraphs lack conciseness: e.g. lines 204-207: "Urine sediment cytology is also frequently used as a complementary test, demonstrating a sensitivity and specificity of 48% (16% for low-grade tumors and 84% for high-grade tumors) and 86%, respectively. However, despite a fairly high sensitivity for high-grade tumors, the sensitivity for low-grade tumors is only 16% [47].” Please revise the text and try combining some phrases or paragraphs in order to obtain a more concise flow.

Response 3: Thank you for this comment. We agree that the sentence presented as an example in the comment needed to be refined for better readability. We have therefore corrected this sentence (page 5, paragraph 3, lines: 235-2240), it now reads as follows: “Urine sediment cytology is also frequently used as a complementary test, demonstrating a sensitivity and specificity of 48% and 86%, respectively. However, despite a fairly high sensitivity for high grade tumors, which is 84%, the sensitivity for low grade tumors is only 16%”.

In addition, in order to achieve a more concise flow and to improve the quality of the text, some cosmetic changes have been made: a) Page 6, paragraph 3, line 244; b) Page 10, paragraph 3.2, lines: 386-388 and 392; c) Page 11, paragraph 3.2, lines: 413, 414, 416; d) Page 16, paragraph 3.3, lines: 467, 475, 476, 481; e) Page 17, paragraph 3.3, lines: 525, 531, 536; f) page 20, paragraph 4.2, line 625. 

Comments 4: Some comments regarding the financial implications of using epigenetic biomarkers and other innovative techniques should be included.

Response 4: We are grateful for this comment. However, we would like to point out that the financial issues related to the practical use of epigenetic biomarkers were addressed in fragments: a) page 6, paragraph 3, lines 260-263: ” Currently, many potential biomarkers based on the modifications in question are already known, but still, few of them have sufficient sensitivity and specificity, especially when it comes to the detection of NMIBC, or have high costs or a complicated procedure for their analysis.”; b) page 7, paragraph 3.1, lines: 306-308: “However, the problems are often too low sensitivity, especially for BC with low malignancy, complicated testing procedures or high costs associated with the use of these assays in practice” c) page 16, paragraph 3.2, lines: 447-450: “None of the available biomarkers has found application in clinical practice yet due to the lack of adequate validation, the high cost of developed tests and diagnostic panels, or complicated and expensive analytical procedures”. Nevertheless, in order to expand the financial aspect of the use of epigenetic biomarkers in the diagnosis and prognosis of BC, the section of the summary found on page 25 and 26, paragraph 5, lines were modified: 707-720: “However, despite the promising results of these studies, epigenetic alterations are still not part of the standard methods for diagnosing and predicting BC, e.g. due to too small study groups, lack of reproducibility of results, lack of adequate validation of the results obtained, the need for appropriate facilities, equipment and highly qualified personnel which can generate high costs. On the other hand, finding an epigenetic biomarker with high sensitivity and specificity could in practice be more cost-efficient than a standardised, expensive cystoscopy. In addition, it is worth noting that finding an effective biomarker to detect BC at an early stage of development could dramatically reduce the costs associated with both the diagnosis and the treatment of the patient itself. Given the above, further research, especially conducted on sufficiently large groups of patients, taking into account the validation of the results obtained, as well as aiming to optimise diagnostic methods and obtain biomarkers that combine all the characteristics of a desirable BC biomarker, seems fully justified”. However, the previous version of this passage (page 26, paragraph 5, lines: 720-727) was deleted. We sincerely hope that the changes we have made will prove sufficiently explanatory of the issues raised in the comment.

All changes made are visible in the file ”jcm3285523_tracking changes”.

Round 2

Reviewer 1 Report

Comments and Suggestions for Authors

Dear Authors,

thank you for carrying out the requested changes in the manuscript
I would only suggest to move the Strength and limitations sections before conclusions

Author Response

Comments 1: thank you for carrying out the requested changes in the manuscript
I would only suggest to move the Strength and limitations sections before conclusions.

Response 1: Thank you very much for this comment. Of course, we agree to reposition the "strength and limitations" section. In line with the comment, this section has been moved (page 25, paragraph 5, lines: 687-702) in front of the section on conclusions (page 25, paragraph 6, lines: 703-729) in new version of the manuscript 

Reviewer 2 Report

Comments and Suggestions for Authors

Thank you so much for answering to all my queries in a satisfactory manner. I believe the added comments and discussions improve the clarity of the manuscript.

I am totally aware that you intended to focus only on epigenetic biomarkers, but before that you created context regarding diagnosis of bladder cancer. From this context, description of other biomarkers and technologies can't be omitted, so thank you for adding comments about them.

Also, I am aware of the previously included comments regarding the costs of epigenetic biomarkers, however, they were quite vague. I appreciated your efforts in adding more clarity to this issue.

Author Response

Comments 1: Thank you so much for answering to all my queries in a satisfactory manner. I believe the added comments and discussions improve the clarity of the manuscript

Response 1: We hope that, having taken into account your comments, the manuscript will present the knowledge it contains in a clear and transparent manner. We are pleased that you find our changes satisfactory.

Comments 2: I am totally aware that you intended to focus only on epigenetic biomarkers, but before that you created context regarding diagnosis of bladder cancer. From this context, description of other biomarkers and technologies can't be omitted, so thank you for adding comments about them.

Response 2: We agree, following your comment, we also feel that the addition of such fragments has enriched our review article.

Comments 3: Also, I am aware of the previously included comments regarding the costs of epigenetic biomarkers, however, they were quite vague. I appreciated your efforts in adding more clarity to this issue.

Response 3: Thank you very much, we have tried to ensure that the changes we made clearly and concisely highlight the issue of the cost of using epigenetic biomarkers in the diagnosis of bladder cancer.